# Metabolomic Markers for the Early Selection of *Coffea canephora* Plants with Desirable Cup Quality Traits

**DOI:** 10.3390/metabo9100214

**Published:** 2019-10-04

**Authors:** Roberto Gamboa-Becerra, María Cecilia Hernández-Hernández, Óscar González-Ríos, Mirna L. Suárez-Quiroz, Eligio Gálvez-Ponce, José Juan Ordaz-Ortiz, Robert Winkler

**Affiliations:** 1Department of Biotechnology and Biochemistry, Cinvestav Unidad Irapuato, Irapuato, Km 9.6 Libramiento Norte Carr. Irapuato-León, Guanajuato 36824, Mexico; 2Red de Biodiversidad y Sistemática, Instituto de Ecología A.C. Xalapa, Veracruz 91070, Mexico; 3Laboratorio de Tecnología del Café, Unidad de Investigación y Desarrollo en Alimentos, TNM/Instituto Tecnológico de Veracruz, Veracruz 91897, Mexico; cecy_mar_19@hotmail.com (M.C.H.-H.); oscargr@itver.edu.mx (Ó.G.-R.); mirna.sq@veracruz.tecnm.mx (M.L.S.-Q.); 4Agroindustrias Unidas de México S.A. de C.V. (AMSA), Bosque de Alisos 45-A 2do Piso, Bosques de las Lomas, Ciudad de México 05120, Mexico; egalvez@ecomtrading.com; 5Laboratorio de Metabolómica y Espectrometría de Masas, Unidad de Genómica Avanzada, CINVESTAV-IPN, Km. 9.6 Libramiento Norte Carr. Irapuato-León, Irapuato 36824, Mexico; jose.ordaz.ortiz@cinvestav.mx

**Keywords:** plant metabolomics, early selection, cup quality, metabolic marker, *Coffea canephora*, metabolome-wide association study (MWAS)

## Abstract

Genetic improvement of coffee plants represents a great challenge for breeders. Conventional breeding takes a too long time for responding timely to market demands, climatic variations and new biological threads. The correlation of genetic markers with the plant phenotype and final product quality is usually poor. Additionally, the creation and use of genetically modified organisms (GMOs) are often legally restricted and rejected by customers that demand natural products. Therefore, we developed a non-targeted metabolomics approach to accelerate conventional breeding. Our main idea was to identify highly heritable metabolites in *Coffea canephora* seedlings, which are linked to coffee cup quality. We employed a maternal half-sibs approach to estimate the metabolites heritability in open-pollinated plants in both leaves and fruits at an early plant development stage. We evaluated the cup quality of roasted beans and correlated highly heritable metabolites with sensory quality traits of the coffee beverage. Our results provide new insights about the heritability of metabolites of *C. canephora* plants. Furthermore, we found strong correlations between highly heritable metabolites and sensory traits of coffee beverage. We revealed metabolites that serve as predictive metabolite markers at an early development stage of coffee plants. Informed decisions can be made on plants of six months old, compared to 3.5 to 5 years using conventional selection methods. The metabolome-wide association study (MWAS) drastically accelerates the selection of *C. canephora* plants with desirable characteristics and represents a novel approach for the focused breeding of crops.

## 1. Introduction

Coffee is one of the most important crops in agriculture around the world since coffee beverages prepared from roasted and ground beans are the second most popular beverage only after tea [1,2]. More than 100 species have been reported belonging to Coffea genus, but *Coffea arabica* and *Coffea canephora* are the most commercialized. Because of their economic importance, these two species have been the main aim of the genetic breeding programs of coffee [3,4]. Traditional genetic breeding strategies for coffee mainly include traditional clones selection, generational selection, selection of pure lines following generational selection, generation of F1 hybrids, interspecific and intraspecific hybridization and selection by backcrossing. The best strategy is chosen according to the behavior of the coffee species, the aim of breeding (disease resistance, quality, yield) and the kind of variety (clone, pure line or variety, hybrid) [5,6,7]. Coffee is a perennial plant, and the first harvest is possible after 3.5 to 5 years. Thus, the selection and breeding of new coffee varieties require at least 25 years, with vast investments of research and development resources.

Last years, priority was given to the improvement of the cup quality of *C. canephora* [8]. This species is considered to have a lower cup quality compared to *C. arabica*. On the other side, *C. canephora* is more resistant against diseases and more tolerant against climatic variations [9,10,11]. Furthermore, the content of caffeine and soluble solids extraction yields are significantly higher, which makes *C. canephora* the principal raw material in the production of instant coffee [4,12,13]. Bean size, soluble solids content, cup quality and cup quality-related compounds (the content of sugars, lipids, caffeine, trigonelline and chlorogenic acids) has been the aim of breeding programs in *C. canephora*. Despite the economic importance of coffee growing and breeding, technological progress is limited compared to other crops [14,15,16,17,18,19]. Therefore, new methodologies for the selection of coffee varieties with desirable agronomical and cup quality characteristics are urgently needed.

Plant phenotypes are tightly related to metabolites. Consequently, the employment of metabolites as predictive markers for complex traits promises to be of high potential. Metabolomics is a powerful tool for the generation of targeted and non-targeted metabolic profiles. The use of advanced statistical analysis provides new possibilities for discovering metabolites, which are predictive for phenotypical characteristics [20,21]. In several crops, such as barley, maize, pine, potato, rice, tomato, wheat and grape [22,23,24,25,26,27,28,29], metabolomics approaches permitted the identification of metabolic markers as accurate predictors of key agronomical traits. The chemical composition of food and beverages also has a direct relation with sensory traits and product quality, such as its nutritional value. Therefore, the raw material used in food productions has a crucial influence on the satisfaction of the consumer. For this reason, we developed new breeding strategies to enable the selection of plant varieties with desirable agronomic and sensory traits.

Metabolome-wide association studies (MWAS) represent a potential tool for the improvement of plants, regardless of the availability of genomic information. Therefore, we applied a metabolomics strategy to identify metabolites predicting sensory traits of coffee beverage in the early development stages of *C. canephora* plants.

## 2. Results and Discussion

### 2.1. Variability of Metabolic Phenotypes and the Relationship Between C. Canephora Parent Plants

Forty families of *C. canephora* were included in this study, comprising a total of 120 coffee plants. Three individuals integrated every family: one parental (mother plant) and two mother half-sibs for every parental (Table 1). The identity of one of the parental was unknown since the coffee plants sibs were reproduced by open-pollinated seeds. To evaluate the natural variation in metabolite levels present in plants of *C. canephora* families, we performed an untargeted metabolomics approach, Ultra Performance Liquid Chromatography-Ion Trap-Mass Spectrometry (UPLC-IT-MS) of aqueous methanolic extracts of fruit and leaf tissues. First, we conducted a hierarchical clustering analysis (HCA) to determine the metabolic variability and relationship between coffee mother plants. We generated two dendrograms depicting the clustering of samples employing the abundances of 755 metabolites detected in fruits and 527 metabolites for leaf tissue. Then, we generated a tanglegram plot in order to compare both phylogenies (Figure 1). Our analyses of phylogeny congruence revealed a high degree (0.81) of entanglement, and comparisons indicated that this was mainly due to the change of position of regions or entire clades in the tree. Naturally, fruit and leaf tissues show very different metabolic profiles. The co-clustering of genetically-related mother plants indicates their metabolic similarity and supports the strong correlation between genotype and metabolic phenotype.

### 2.2. Selection of Highly Heritable Metabolites in Coffee Fruits and Leaves of C. Canephora Families

In coffee fruits, we detected 755 metabolites, which were present in all 120 samples, whereas a minor quantity of metabolites was detected in leaves (527 metabolites). Metabolites in plants are tightly associated with phenotypic characteristics and represent a high diversity at a quantitative and qualitative level within plant species. The application of untargeted metabolomics analysis enables the detection of a wide range of metabolite classes in plants and support the study of aspects regulating biochemical variation such as environmental and genetic control and how these compounds influence complex traits [20,30,31].

To evaluate the variability in metabolite levels and determine the contribution of the genetic factor, we determined the abundance of every metabolite and estimated the metabolic heritability in the narrow sense by offspring-parents regression. Heritability levels for traits range from 0 to 1. Scores less than 0.2 are categorized as low heritability, scores between 0.2 and 0.39 are considered as moderate heritability and high heritability corresponds to values above 0.4. In fruits, our results revealed a high percentage of metabolites (mass peaks) with high heritability (59.6%, 450 metabolites), while 97 and 208 metabolites exhibited low and moderate heritability levels, respectively (Figure 2A). In contrast, for metabolites heritability estimated in leaf tissues, we detected 22.01% (116 metabolites) with high heritability values, 125 metabolites of moderate heritability (23.72%) and 286 metabolites (54.27%) were classified as low heritability metabolites (Figure 2B). Highly heritable metabolites identified from mother plants to F1 progeny: 450 in fruits and 116 in leaves were considered for the posterior analysis of correlation with sensory traits of coffee beverage. Our results demonstrated that for a considerable number of metabolites, a high percentage of the observed quantitative variation could be explained by genetic factors, as demonstrated by their high heritability levels. A study of metabolic diversity by using metabolomics approaches permits to evaluate natural variation occurring within and between species [32]. Our findings are relevant for the knowledge of the variability naturally occurring in the levels of metabolites of coffee tissues and the contribution of genetic factors. Studies of heritability of qualitative and quantitative characters of the coffee crop have been the basis for the genetic improvement of quantitative traits such as yielding, disease resistance, adaptability to climatic and soil conditions, bean physical quality and sensory traits of coffee beverage [19,33,34,35,36,37]. Our highlights, together with association studies between metabolites and coffee traits, could be of great importance to support the classical breeding of coffee plants with desirable traits, especially those that are associated with the natural variation of metabolite composition and levels.

### 2.3. Metabolites of Roasted Coffee Beans Linked to Sensory Traits of Coffee Beverage

#### 2.3.1. Tasting and Correlation Between Sensory Attributes of Coffee Beverage

For investigating the relationships between sensory traits of coffee beverage, a panel of 11 trained cup-testers evaluated 21 roasted coffee bean samples. Eleven samples randomly selected corresponded to *C. canephora* mother plants (M480–M495) and eight *C. arabica* and *C. canephora* samples were used as reference samples. Ten sensory attributes of coffee beverage were tested, including aromatic quality, aromatic intensity, astringency, body, flavor, after taste, bitterness, acidity, sourness and global preference, using a scale ranging of 0 to 5 points (Table 2). First, we performed a correlation analysis between sensory attributes (Figure 3). Based on a Spearman correlation analysis, the aromatic intensity attribute only showed a positive correlation with aromatic quality, with a Spearman correlation coefficient (SCC) of 0.48 and statistical significance (*p* < 0.05). Aromatic quality was positively correlated with eight out of nine attributes with the exception of acidity (SCC: −0.55; *p* < 0.01) and sourness (SCC: −0.62; *p* < 0.01) for which negative correlation was observed. We obtained a similar correlation pattern for flavor, astringency and aftertaste. The latter demonstrated positive correlation with statistical significance for most of the attributes except for acidity and sourness. Interestingly, for acidity, we detected negative correlation for six attributes (aromatic quality, flavor, aftertaste, bitterness, body and global preference) and only for the sourness attribute, we found a high Spearman correlation coefficient of 0.87 (*p* < 0.001). Global preference represents the rating of the overall perception of all the organoleptic characters of coffee beverage. This attribute revealed positive correlation for aromatic quality, flavor, astringency, aftertaste, bitterness and body. However, we found a high negative correlation for acidity and sourness attributes (SCC: −0.85 and −0.8; *p* < 0.001, respectively). The close relationship between acidity and sourness and the correlation between all the sensory traits assessed is illustrated in a dendrogram of correlation distances (Figure 4). Altogether, our results demonstrated the existence of relationships between the majority of the sensory traits, and suggest that attributes do not only affect coffee quality individually.

#### 2.3.2. Selection of Marker Metabolites in Roasted Coffee Beans Linked to Sensory Traits

Coffee cup quality is determined by diverse factors, such as the coffee plant genotype, environmental factors (geographical origin, climate, altitude, temperature, percentage of crop shade, soil nutrient capacity, fertilization), agricultural practices, post-harvest processing and grain storage, as well as the basic chemical composition of coffee bean. A wide range of compounds contributes to the coffee cup quality, including volatile and non-volatile compounds, which determine the organoleptic quality of coffee beverage [38,39,40,41,42].

Non-volatile compounds of coffee roasted beans include carbohydrates, sucrose, free sugars, cell wall polysaccharides, lignin, nitrogen compounds (proteins, free amino acids, caffeine, trigonelline), lipids, minerals (potassium, phosphorus), organic acids, esters (chlorogenic acids, aliphatic acids, quinic acid derivates) and melanoidins [19,35]. To identify metabolite markers of coffee cup quality present in roasted coffee beans, we analyzed aqueous extracts of roasted and ground beans by UPLC-Electrospray (ESI)-MS. We detected 737 metabolites that were present in all 21 samples of *C. canephora* and *C. arabica* samples included for this analysis. We performed a Spearman correlation analysis employing the levels of detected metabolites versus the scores of each of the 10 sensory traits of coffee beverage before mentioned. We detected 392 correlations that fulfilled those characteristics, involving 174 metabolites and eight sensory attributes of coffee beverage, with SCC values ranging from 0.5 to 0.813 for positive correlations and −0.5 to −0.849 for negative correlations (see Appendix A). The results indicated no metabolites (with statistical significance) linked to aromatic intensity and astringency attributes, whereas for all the other sensory attributes, at least 21 correlations were found. To identify the metabolites, we compared high-resolution accurate mass and MS data by using the programs XCMS, MSDIAL and SpiderMass. We putatively identified 91 (52.3%) of the metabolites (Appendix A), which were classified in 11 class compounds, comprising alkaloids, carbohydrates, carotenoids, chlorogenic acids, fatty acids, flavonoids, hydrocarbons, lipids, organic acids and terpenoids (Figure 5). Several metabolites of these class compounds have been identified as biochemical compounds playing a role in roasting chemistry. For example, carbohydrates and free sugars suffer pyrolysis, thermal and Maillard (with proteins and amino acids) reactions generate aroma precursors such as furanes, carboxylic acids, pyridines, pyrroles, pyrazines, thiazoles and ketones. Ferulic, caffeic and quinic acids from chlorogenic acids were degraded to produce phenols, and lipids through oxidation were transformed to alcohols, ketones and aldehydes [2,43,44,45,46].

Interestingly, we detected a high number of lipids influencing the sensory attributes of coffee beverage. Fifty-two metabolites (114 associations) were annotated as lipids comprising four classes: glycerolipids (GL), glycerophospholipids (GP), prenol lipids (PR), sphingolipids (SP) and sterol lipids (ST) (Appendix A). Lipids of roasted coffee beans enhance the organoleptic properties of coffee beverage such as flavor, color and foam. Lipid composition of beans is made mainly of triacylglycerides (75%) and terpene esters (14%), free fatty acids (1%), free sterols (1.5%), sterol esters (1%) and phospholipids (PLs) which are present in lower levels [47]. We detected a phospholipids group containing SP and GP classes. SPs were associated with acidity, aftertaste, bitterness, global preference and sourness, whereas GPs were negatively or positively linked to all the evaluated attributes (Appendix A). GLs were mainly associated with acidity and body attributes, for which the majority of lipids were negatively associated (SCC = −0.50 to −0.584). Although there is much information about compounds present in coffee roasted beans or coffee beverage, limited information exists about the individual or specific chemical components that are directly linked to organoleptic characteristics of coffee beverage. Identification of metabolites linked to several sensory traits is crucial because of their value as marker metabolites for specific coffee sensory traits. In our study, we successfully identified some metabolites that were negatively or positively correlated to eight out of 10 assessed sensory attributes.

Figure 6 shows association levels of sensory traits with metabolites putatively identified as quercetin 4’-glucoside and trigonelline representing two examples of marker metabolites presented as critical components which confer specific coffee sensory properties. Quercetin 4´-glucoside is an organic compound that belongs to flavonoid o-glycosides, which contain a carbohydrate moiety linked to the 2-phenylchromen-4-one flavonoid backbone by O-glycosidic bond. Quercetin 4´glucoside was found to have a positive correlation with acidity and sourness (SCC: 0.56 and 0.64; p:0.0089 and 0.0017, respectively), whereas a negative correlation was observed for bitterness, flavor, aromatic quality, aftertaste, body and global preference attributes (Figure 6). Some glycosides have been determined as marker metabolites influencing the sensory profile of the coffee beverage. Mozambioside was reported as an Arabica-specific bitter-tasting furokauran glucoside detected in roasted coffee beans and beverage [48], whereas 3-methylbutanoyl glycosides were identified as markers for the quality of coffee flavors [49]. Trigonelline was linked to seven sensory traits, but a positive correlation was observed only for acidity, whereas for aromatic quality, bitterness, body, aftertaste, flavor and global preference, we detected a negative association. It is well-known that 50–80% of trigonelline is degraded by pyrolysis after the roasting process and contributes to the formation of other compounds such as nicotinic acid, pyridine, 3-methyl-pyridine, the methyl ester of nicotinic acid and 1-methylpyridinium [50,51]. All these products of trigonelline degradation finally influence organoleptic features of coffee beverage. Thus, the trigonelline content in roasted coffee beans represents a predictive marker for more than one sensory trait.

### 2.4. Highly Heritable Metabolites in Fruits Linked to Coffee Cup Quality

Some aspects, such as ploidy incompatibility between *C. arabica* and *C. canephora*, as well as the time that elapses from seed germination until the first harvest (3.5 to 5 years), represent a disadvantage for traditional breeding methods. For these reasons, obtaining an improved coffee variety might take up to 30 years [52,53]. Responding to the aforementioned, we performed a correlation analysis, which included the 450 metabolites mentioned, identified as highly heritable metabolites and scores of 10 sensory traits of coffee beverage. We detected 228 correlations involving 158 metabolites that were associated to at least one sensory trait and fulfilled the following characteristics: values of SCC > 0.5 for positive correlation and < −0.5 for negative correlation, and statistical significance (*p* < 0.05). 141 associations (61.8%) were negative correlations (SCC: −0.666 to −0.96) whereas 87 associations were detected as positive correlations (SCC: 0.66 to 0.945) (See Appendix A). Aromatic quality, aromatic intensity and bitterness displayed the highest number of associations to highly heritable metabolites, 39, 29 and 29, respectively. On average, we detected 1.5 significant associations per metabolite feature. In Table 3, we list the top metabolites of coffee fruits with optimal characteristics, which explicated high predictive value for sensory traits of coffee beverage and for selection of plants producing coffee beans with desirable sensory characteristics. In the top metabolites list, we included 2,4-decadienoyl-CoA (*m/z* 940.2271), a metabolite constituted by unsaturated fatty acids with trans double bonds and which is degraded into acetyl-CoA by a beta-oxidation. High levels of this fatty acyl thioester of coffee fruits were positively associated with aftertaste (heritability: 0.881; SCC: 0.754).

The mass peak *m*/*z* 791.4983, putatively identified as nonaprenyl diphosphate, demonstrated a high heritability level (0.984) and positive correlation with global preference (SCC = 0.728). We identified two compounds that belong to polyphenols/flavonoids class: 2,3-trans-proanthocyanidin with high predictive value for astringency (heritability: 0.957; SCC: 0.833) and delphinidin 3-O-glucosyl-5-O-caffeoylglucoside which presented a positive correlation with body sensory trait (heritability: 0.690; SCC: 0.833). Six unidentified mass peaks (*m*/*z* 963.4134, *m*/*z* 860.5079, *m*/*z* 536.0019, *m*/*z* 734.4907, *m*/*z* 598.5406, *m*/*z* 521.5618) were associated to acidity, aromatic intensity, aromatic quality, bitterness, flavor and sourness, respectively. Improvement of organoleptic traits in coffee requires knowledge about the chemical composition of tissues of coffee plants, which contribute to the final product: roasted beans and coffee beverage [53]. The presented markers assist in the improvement and manipulation of sensory traits of coffee beverages. In the past, quality improvement of *C. canephora* has been performed for bean size, extractable soluble solids, sugars, caffeine, trigonelline, total lipids content and chlorogenic acids [8]. Here, we reported the first study that combined analysis of high throughput metabolite heritability in combination with a correlation analysis of sensory traits of coffee beverage. In other plant species, such as *Arabidopsis* [30], maize [28], tomato [54,55,56] and raspberry [57], the application of metabolomics together with complementary tools, such as genome-wide selection, has permitted the study of metabolite heritability, the elucidation of factors controlling biochemical variability and the purpose for modification of chemical composition by classical breeding. We reported a considerable number of metabolites linked to sensory traits, which supports the early selection of *C. canephora* plants.

### 2.5. Highly Heritable Metabolites in Leaf Linked to Coffee Cup Quality

To evaluate the link between highly heritable metabolites that have been detected before in coffee leaves and the values of 10 sensory traits of coffee beverage assessed by the trained cup testers, we performed a correlation analysis. We found 75 associations between 45 metabolites, which were linked at least with one sensory attribute of coffee beverage (Appendix A). The average of significant associations per mass peak was 1.7, slightly higher than values found for fruits (1.5). Forty-six associations, corresponding to 61.3%, were negative correlations (SCC: −0.663 to −0.9746), and the remaining 38.7% of associations were positive correlations with values of SCC between 0.666 and 0.866) (Appendix A). Mass peaks *m/z* 251.4295, *m/z* 387.3897, *m/z* 473.6946, *m/z* 517.7342, *m/z* 587.1469, *m/z* 695.3553 and *m/z* 805.7835 are interesting because all presented that they were associated at least with three coffee sensory traits of coffee beverage. This fact is of great importance, because we can evaluate multiple sensory traits simultaneously by screening the levels of a single metabolite. In Table 3, we listed the top metabolites purposed as excellent candidates for early selection of plants with desirable coffee sensory traits due to their outstanding characteristics of high heritability and strong positive correlation level to cup quality traits. 6-deoxoteasterone (*m/z* 473.3339), involved in brassinosteroid biosynthesis I in coffee plants, was linked to three sensory traits: acidity, sourness and bitterness. For the last one, a negative correlation (SCC = −0.857) was detected. It was included in the list, because for bitterness, only negative associations were observed. We found a similar situation for the ‘body’ attribute: 4,4-dimethylzymosterol (*m/z* 463.3451) was negatively linked to this trait. Two lipids, PA(16:0/18:2) (a glycerophosphate: GP) and MGDG(23:0/26:0) (a glycosyldiradylglycerol: GL05) were found in coffee leaves and presented high predictive values for aftertaste and astringency. Levels of mirystic acid, a fatty acid (FA) resulted as a suitable predictor for aromatic quality and flavor (heritability: 0.994; SCC: 0.766 and 0.672, respectively). Global preference was strongly associated with levels of gibberellin A28, with heritability levels of 0.996 and an SCC of 0.677. A level distribution analysis of all that metabolites in the 40 *C. canephora* parent plants (Figure 7) highlights individuals which can be subject for the breeding of plants with specific sensory characteristics. Therefore, we provided new knowledge about the inheritance of coffee leaf metabolites, but also information about the link between metabolite levels in the leaf and complex agronomic traits such as coffee cup quality. Remarkably, the use of highly heritable metabolite markers for coffee breeding in 6-month-old plants, instead of waiting for 3.5–5 years until the first coffee fruit harvest, opens perspectives to reduce the time for generation of coffee varieties with desirable sensory traits. Nevertheless, it is essential to emphasize that the chemical composition and sensory properties of coffee are not only determined by genetic factors, but also by cultivation conditions, plant condition, cultivation practices, harvest and post-harvest processes [19,39,58,59]. Therefore, a strategy for improving traits of interest must integrate all those non-genetic factors as well. Selecting plants with the proposed strategy would comprise: a) screening of a parent plant population using marker metabolites, b) selection of plants with desirable characteristics based in metabolite levels and c) plant propagation.

## 3. Materials and Methods

### 3.1. Heritability Analysis

#### 3.1.1. Leaves and Fruits Sampling

Samples for heritability analysis were collected from an experimental farm in Tapachula, Chiapas, Mexico. We designed a maternal half-sibs approach in order to estimate the heritability of metabolites in both tissues, leaves and fruits. In this study, 120 *C. canephora* plants were included, consisting of 40 mother plants and their F1 progeny (two sibs for every mother plant), which was reproduced by open-pollinated seeds. Leaves of all plants (mothers and sibs) were sampled at 6 months of age for F1 plants and, three years later in a second sampling (at first fruit harvest of F1 plants), completely mature fruits of both mother plants and sibs were obtained.

#### 3.1.2. Leaves and Fruits Tissue Metabolite Extraction

Approximately two grams of freeze-dried tissue were ground in a Mixer Mill MM 301 (Retsch^®^) at 30 Hz for 15 s and sieved with a mesh for <300 µm particle size. Then, 35 mg of every sample was extracted twice in 450 µL of solvent (75% methanol/24.85% water, 0.15% formic acid), 25 min in an ultrasound bath at room temperature. After centrifugation at 15,870× *g* for 5 min at room temperature, the supernatants of first and second extraction were collected in the same 1.5 mL tube and filtered using a 0.2 µm membrane. These fractions were subjected to UPLC-MS analysis.

#### 3.1.3. Estimation of Metabolites Heritability

Narrow sense heritability of metabolites was calculated through offspring-parents regression, which is suitable for cross-fertilizing crops, and when the estimation of heritability is performed for half-sib families regressed on a single parent. Measures corresponded to intensity levels of every metabolite detected. The regression coefficient (Y_i_), which measures the rate between covariance of parents to offspring covariance, with its respective standard error was calculated as follows: Y_i_ = β_0_ + β_1_X_i_ + ε_i_, where: *Y_i_* = Average measures of offspring of the X families; β_0_ = intercept; β_1_ = regression coefficient; X_i_ = Measurement of only one parent (mother plant) per family; ε_i_ = Standard error. The slope of the regression line (β_1_) represents an estimate of narrow-sense heritability (h^2^), which depends mainly on additive genetic values. Narrow-sense heritability (h^2^) is very important for plant breeding programs because artificial selection mainly depends on additive genetic variance.

### 3.2. UPLC-ESI-MS Analysis

Metabolite analysis was performed employing an UPLC-ESI-MS (Ultra Performance Liquid Chromatography Electrospray Mass Spectrometry) system consisting of an Accela UPLC coupled to LCQ Fleet Ion Trap (Thermo Finnigan, San José, CA, USA). Separation of compounds was performed on a Hypersil Gold C18 column (50 × 2.1 mm, 1.9 µm particle size). In the UPLC system, mobile phase A consisted of 0.1% FA in water and solvent B was MetOH with 0.1% FA. For leaves analysis, the solvent gradient was as follows: 8% B, 0–0.5 min; 8–65% B, 0.5–6.5 min; 65%–100% B, 6.5–20 min; 100% B during 1 min (20–21 min); 0.5 min for comeback to re-equilibration conditions and finally, 8% B during 3.5 min (21.5–25 min). The gradient standardized for fruit tissues was: 3% B during 1 min; 3–50%, 1–5 min; 50–73% B, 5–5.5 min; 73–100% B, 5.5–19.3 min; 100% B during 1min; 100% B–3% B, 20.3–21.5; and 3.5 min 3% B for column re-equilibration. For roasted coffee analysis, the solvent gradient was: 0–1 min, 2% B; 1–6 min, 2–66% B; 6–12 min, 66–88% B; 12–19 min, 88–93% B; 93–97% B in 1 min; 97–100% B, 20–25 min; from 25 to 27 min was maintained at 100% B; 27–27.5 min, 100% B–98% A and finally, column re-equilibration for 3.5 min with 2% B. The column oven temperature was always maintained at 40 °C, and the flow rate was 550 µL/min. Spectra were acquired in a range of 50–1000 *m/z.* The analysis was made for separated in both positive and negative mode. The scan time was 500 ms (3 micro-scans). The ESI source parameters employed for positive mode were: capillary temperature 330 °C; capillary voltage 35 V; spray voltage 4 kV; Tube lens 80 V; nitrogen sheath gas 30 arbitrary units (AU); auxiliary gas 15 AU. For negative mode, the capillary voltage was −45 V and −123.38 V for tube lens. For the MS analyses, the collision energy (CID) was normalized to 35.

### 3.3. Spectra Data Analysis and Putative Identification of Metabolites

Raw mass spectrometry data processing for feature detection was performed in MZmine 2.37 software [60]. A base-line correction filter was applied and peak detection was performed with GridMass, with the following parameters: 0.32 *m/z* tolerance, 0.1 to 0.8 min width and a minimum peak height of 500. Isotopic peaks grouper was applied, and then, RANSAC (Random Sample Consensus) alignment was conducted with 0.35 *m/z* tolerance and 1.3 min of retention time. Finally, Gap filling (same RT and *m/z*) and duplicate peak filter were applied with 0.35 *m/z* tolerance. Intensities of features (*m/z* values) were employed for all statistical data analysis. Additionally, pools of samples were analyzed by high resolution-MS comprising a QTOF Synapt G1 (Waters, UK) in order to get accurate mass data. Three strategies were used for the putative identification of metabolites: (1) analysis of high-resolution MS data in XCMS [61] by matching with the METLIN database; (2) matching of accurate masses list by using a *Coffea sp*. in-house generated database with SpiderMass software [62]; (3) comparison of tandem MS data from LCQ Fleet Ion Trap with METLIN database, which was conducted in XCMS Online.

### 3.4. Cup Quality-Related Metabolite Analysis

#### 3.4.1. Coffee Beans Processing and Sensory Analysis of Coffee Beverage

About five kilograms of fully ripe fruits were harvested for every plant included in the analysis. Fruit samples were subjected to post-harvest drying processing to obtain green coffee beans free of peel and parchment with moisture content about 12%. Green coffee beans of *C. canephora* and *C. arabica* (as reference samples) were medium roasted during 7–8 min according to SCAA specifications (Specialty Coffee Association of America 2015). The degree of roasting was standardized to a value of 55 (medium roast) in a colorimetric Agtron scale developed by SCAA. The coffee beverage was prepared using 50 g of roasted beans ground to medium size, and 1000 mL of boiled water (97 °C). After cool down to 50 °C, the beverage of all roasted coffee samples were tested by a trained panel of 11 trained cup-testers, who evaluated 10 attributes related to coffee cup quality: aromatic quality, aromatic intensity, astringency, body, flavor, after taste, bitterness, acidity, sourness and global preference, employing a scale ranging of six points (0 to 5) where 0 = no presence and 5 represents the higher intensity of the attribute.

#### 3.4.2. Roasted Beans Metabolite Extraction

35 mg of ground roasted beans were extracted twice in 450 µL of 95 °C water in an ultrasound bath for 25 min. Samples were centrifuged at 15,870× *g* for 5 min at room temperature, and the supernatant was filtered through a 0.2 µm membrane and employed for UPLC-MS analysis.

## 4. Conclusions

Our study represents a comprehensive analysis of the heritability of metabolites in *Coffea canephora* plants and reports predictive metabolic markers for the coffee cup quality markers. We proved the feasibility to use metabolic profiles for the early selection of coffee plants in breeding projects and to speed up the development of new coffee varieties drastically. Compared to conventional breeding strategies, a ‘metabolome-wide association study’ (MWAS) permits a shorter time-to-market, and therefore, enables a more agile development of commercial products. Moreover, the creation of more resistant plant varieties with adequate cup quality is accelerated. The latter might become critical for responding to environmental challenges due to the climatic change.

MWAS is applicable to plant collections of any origin: natural varieties, conventional breeding and genetically modified organisms (GMOs). The analytical and statistical methods are highly scalable. Thus, MWAS has high potential in the breeding of crops of plants of agricultural or industrial interest. In further studies, we will test metabolic markers for sensory quality with *C. arabica* and study other complex traits, such as fungal resistance.

## Figures and Tables

**Figure 1 metabolites-09-00214-f001:**
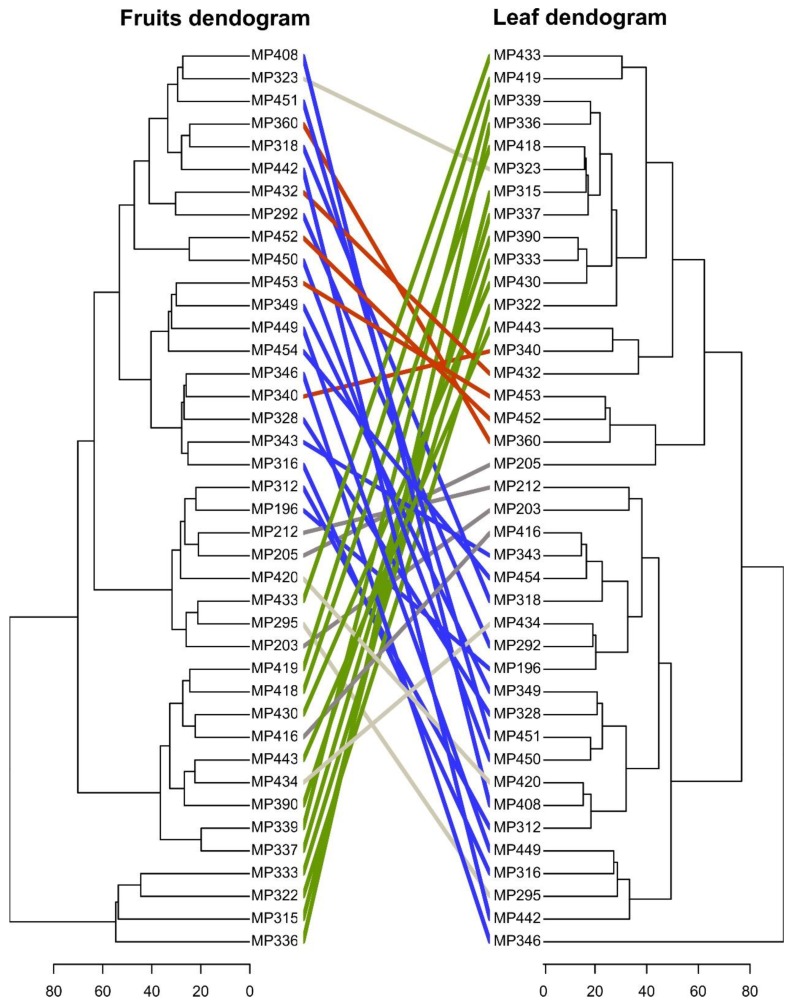
The tanglegram compares the hierarchical clustering of mother plants employing fruits versus leaves metabolite profiles. Individual dendrograms were calculated using hierarchical clustering with the ward.D2 method. The signal intensity of every peak mass used for generation of dendrograms consisted of the average of three technical replicates.

**Figure 2 metabolites-09-00214-f002:**
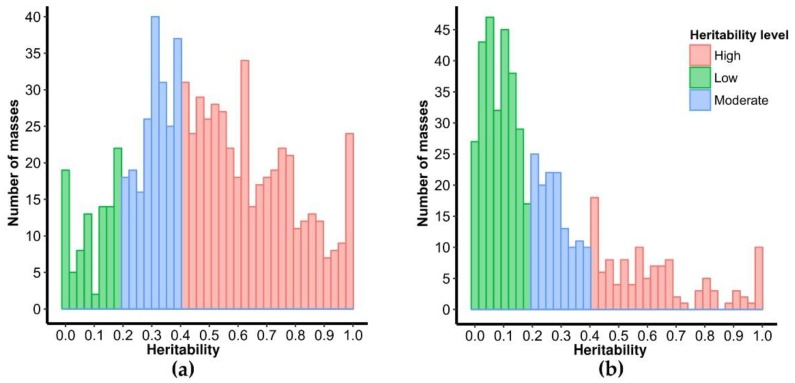
Frequency distribution of heritability of mass peaks detected in fruits (**a**) and leaves (**b**). The histograms demonstrate the distribution of the heritability level. Colors depict values according to low, moderate and high heritability levels.

**Figure 3 metabolites-09-00214-f003:**
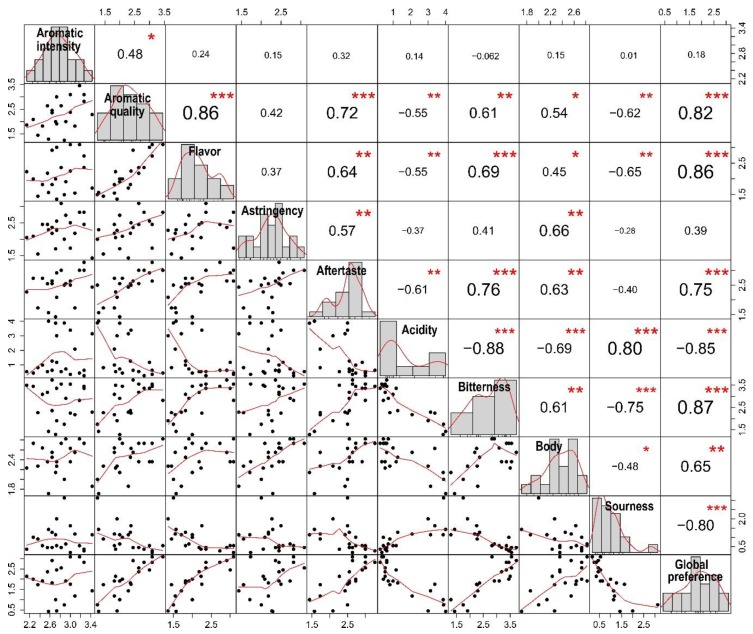
The correlation plot shows histograms and density distribution of cup quality attributes of coffee beverage. The Spearman correlation coefficient (SCC) between sensory traits of the coffee beverage is depicted. Data with statistical significance levels are indicated with asterisks: *p*-values ≤ 0.05 *, ≤0.01 ** and ≤0.001 ***.

**Figure 4 metabolites-09-00214-f004:**
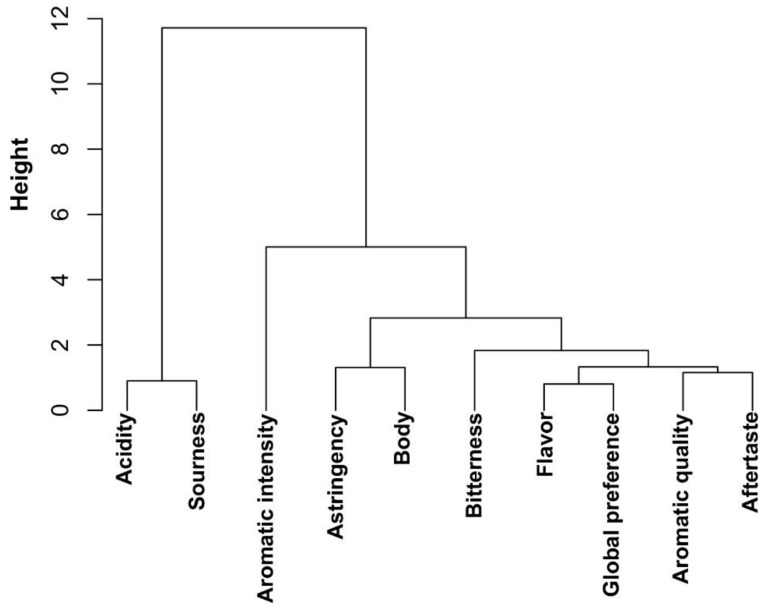
Cluster dendrogram of correlation between cup quality attributes of coffee beverage. Hierarchical clustering was calculated with the ward.D2 method; correlation distances were calculated by employing the scores of every sensory trait determined by the trained cup-tester panel.

**Figure 5 metabolites-09-00214-f005:**
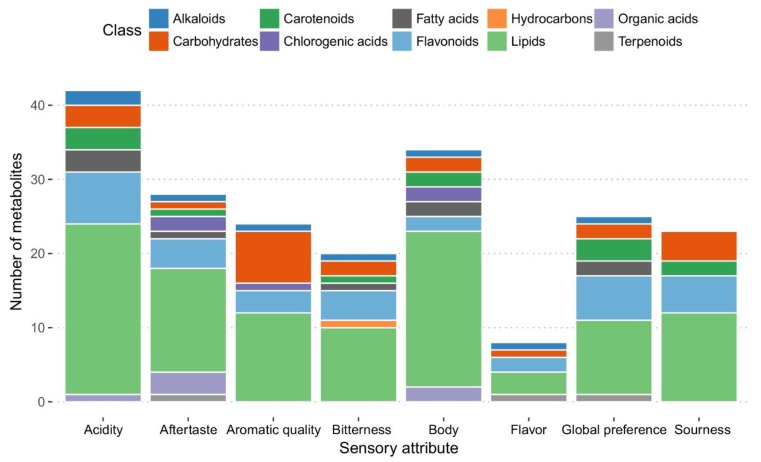
Classification of roasted bean metabolites linked to sensory traits of coffee beverage. Only putatively identified metabolites were included for compound classification by using KEGG and Pubchem tools.

**Figure 6 metabolites-09-00214-f006:**
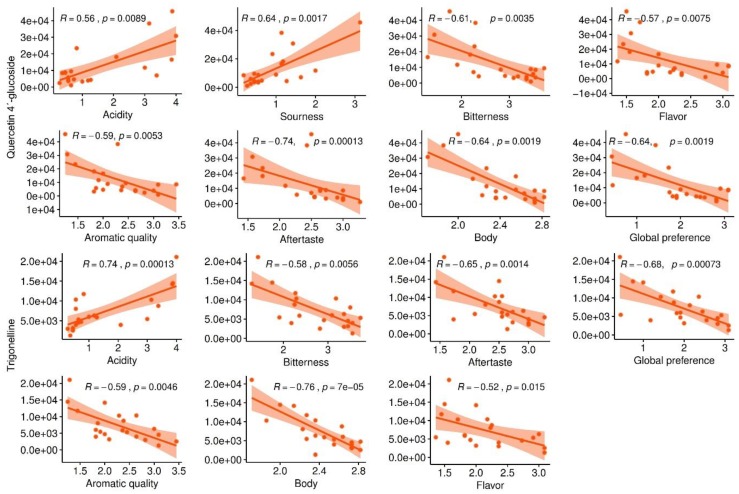
Metabolites of roasted coffee beans linked to multiple sensory traits. The figure shows a significant Pearson correlation (positive and negative) of quercetin 4´glucoside and trigonelline with multiple sensory traits of coffee beverage.

**Figure 7 metabolites-09-00214-f007:**
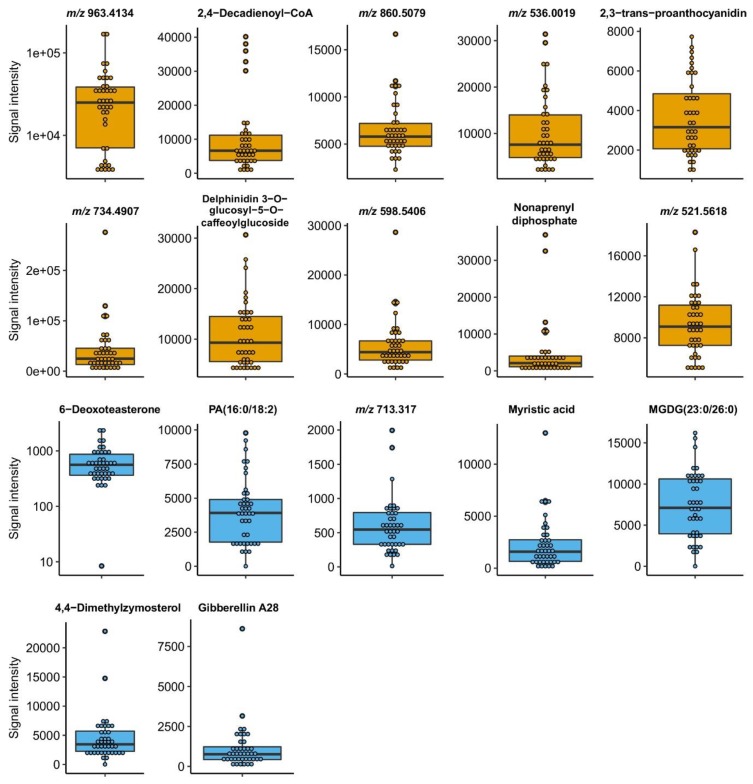
Distribution of abundances of top metabolites in parent *C. canephora* plant tissues. Level distribution across the 40 *C. canephora* parent plants are depicted for the metabolites listed in Table 3. Yellow-colored boxplots indicate metabolites found in fruits and blue color represents metabolites from the coffee leaf.

**Table 1 metabolites-09-00214-t001:** *C. Canephora* families included in heritability analysis.

Number	Mother Plant ID	Sib 1	Sib 2	Number	Mother Plant ID	Sib 1	Sib 2
1	MP196	196-1	196-3	21	MP349	349-2	349-3
2	MP203	203-1	203-2	22	MP360	360-3	360-4
3	MP205	205-1	205-2	23	MP390	390-1	390-3
4	MP212	212-1	212-2	24	MP408	408-1	408-4
5	MP292	292-3	292-4	25	MP416	416-1	416-3
6	MP295	295-2	295-3	26	MP418	418-3	418-4
7	MP312	312-3	312-4	27	MP419	419-2	419-4
8	MP315	315-1	315-2	28	MP420	420-1	420-4
9	MP316	316-1	316-2	29	MP430	430-2	430-3
10	MP318	318-2	318-3	30	MP432	432-1	432-2
11	MP322	322-3	322-4	31	MP433	433-2	433-3
12	MP323	323-1	323-4	32	MP434	434-2	434-4
13	MP328	328-3	328-4	33	MP442	442-2	442-3
14	MP333	333-1	333-4	34	MP443	443-3	443-4
15	MP336	336-1	336-2	35	MP449	449-1	449-3
16	MP337	337-3	337-4	36	MP450	450-3	450-4
17	MP339	339-1	339-2	37	MP451	451-1	451-4
18	MP340	340-1	340-3	38	MP452	452-2	452-3
19	MP343	343-1	343-3	39	MP453	453-2	453-3
20	MP346	346-1	346-4	40	MP454	454-2	454-4

**Table 2 metabolites-09-00214-t002:** Sensory profile of coffee beverages.

Sample	Aromatic Intensity	Aromatic Quality	Flavor	Astringency	Aftertaste	Acidity	Bitterness	Body	Sourness	Global Preference
MP302	3.27 ± 0.52	3.09 ± 0.33	2.73 ± 0.39	2.45 ± 0.49	3.27 ± 0.39	0.64 ± 0.46	3.45 ± 0.49	2.73 ± 0.52	0.27 ± 0.39	2.82 ± 0.29
MP307	2.14 ± 0.33	1.86 ± 0.47	2.23 ± 0.41	2.05 ± 0.33	2.27 ± 0.38	0.55 ± 0.33	3.50 ± 0.54	2.23 ± 0.61	0.36 ± 0.46	2.14 ± 0.24
MP309	3.18 ± 0.44	3.45 ± 0.49	3.09 ± 0.33	2.82 ± 0.29	3.00 ± 0.18	0.45 ± 0.49	2.82 ± 0.29	2.82 ± 0.29	0.55 ± 0.50	3.09 ± 0.16
MP324	2.55 ± 0.49	2.64 ± 0.46	2.18 ± 0.29	2.45 ± 0.49	2.55 ± 0.67	0.55 ± 0.49	3.18 ± 0.47	2.36 ± 0.46	0.45 ± 0.49	2.36 ± 0.46
MP316	2.64 ± 0.46	2.45 ± 0.49	2.91 ± 0.33	2.36 ± 0.46	2.64 ± 0.57	0.64 ± 0.46	3.73 ± 0.39	2.55 ± 0.49	0.45 ± 0.49	2.91 ± 0.16
MP318	2.82 ± 0.29	2.64 ± 0.46	2.36 ± 0.46	3.09 ± 0.16	3.00 ± 0.18	0.55 ± 0.49	3.55 ± 0.49	2.73 ± 0.39	0.45 ± 0.49	2.55 ± 0.49
MP323	3.00 ± 0.18	3.09 ± 0.16	3.09 ± 0.33	1.73 ± 0.39	2.64 ± 0.46	0.36 ± 0.46	3.55 ± 0.49	2.36 ± 0.57	0.18 ± 0.29	3.09 ± 0.33
MP346	2.27 ± 0.39	2.09 ± 0.16	2.00 ± 0.36	2.09 ± 0.16	2.73 ± 0.39	0.45 ± 0.50	3.45 ± 0.49	2.73 ± 0.39	1.00 ± 0.18	2.00 ± 0.18
MP360	2.64 ± 0.52	2.36 ± 0.46	1.82 ± 0.29	2.18 ± 0.29	2.55 ± 0.49	1.27 ± 1.07	2.36 ± 0.82	2.45 ± 0.77	1.27 ± 0.39	1.82 ± 0.29
MP408	3.27 ± 0.39	2.82 ± 0.29	2.36 ± 0.46	2.82 ± 0.29	3.00 ± 0.36	0.27 ± 0.39	3.36 ± 0.46	2.73 ± 0.39	0.36 ± 0.36	2.82 ± 0.29
MP420	2.73 ± 0.39	3.00 ± 0.18	3.00 ± 0.36	2.55 ± 0.49	2.91 ± 0.33	1.18 ± 0.29	3.36 ± 0.46	2.36 ± 0.46	0.55 ± 0.49	2.45 ± 0.49
MP453	2.45 ± 0.49	1.82 ± 0.29	1.82 ± 0.47	2.45 ± 0.49	2.73 ± 0.39	1.00 ± 0.18	3.09 ± 0.16	2.64 ± 0.46	0.55 ± 0.49	1.91 ± 0.16
MP454	2.91 ± 0.33	2.00 ± 0.18	1.91 ± 0.17	2.73 ± 0.39	2.55 ± 0.49	0.73 ± 0.4	2.91 ± 0.33	2.82 ± 0.29	0.64 ± 0.46	1.91 ± 0.16
M495	2.36 ± 0.46	1.45 ± 0.49	1.45 ± 0.49	1.55 ± 0.59	1.73± 0.39	0.29 ± 0.74	2.27 ± 0.39	2.27 ± 0.39	0.91 ± 0.33	1.73 ± 0.52
M498	2.73 ± 0.39	1.91 ± 0.16	1.36 ± 0.46	2.27 ± 0.39	2.09 ± 0.16	3.00 ± 0.18	1.91 ± 0.33	2.27 ± 0.40	2.00 ± 0.36	0.45 ± 0.49
M499	3.09 ± 0.33	1.82 ± 0.29	1.55 ± 0.49	2.18 ± 0.29	1.73 ± 0.39	2.09 ± 0.49	2.18 ± 0.29	2.55 ± 0.49	1.18 ± 0.30	1.18 ± 0.29
M500	3.00 ± 0.25	2.38 ± 0.46	2.25 ± 0.38	2.50 ± 0.50	2.50 ± 0.50	3.38 ± 0.46	2.25 ± 0.37	2.63 ± 0.46	1.63 ± 0.47	1.75 ± 0.37
M502	3.43 ± 0.48	2.29 ± 0.40	1.71 ± 0.40	1.43 ± 0.48	2.43 ± 0.48	3.14 ± 0.24	2.29 ± 0.40	1.86 ± 0.24	1.14 ± 0.24	1.43 ± 0.49
M506	2.88 ± 0.22	1.25 ± 0.38	1.50 ± 0.50	2.00 ± 0.25	2.50 ± 0.50	3.88 ± 0.44	1.75 ± 0.38	2.00 ± 0.25	3.13 ± 0.44	0.75 ± 0.38
M507	2.57 ± 0.49	1.29 ± 0.41	1.57 ± 0.49	2.29 ± 0.69	1.57 ± 0.49	4.00 ± 0.57	1.43 ± 0.61	1.71 ± 0.41	1.43 ± 0.49	0.43 ± 0.49
M508	2.71 ± 0.26	2.18 ± 0.29	2.00 ± 0.18	1.71 ± 0.40	1.43 ± 0.49	3.86 ± 0.24	1.29 ± 0.40	2.14 ± 0.24	1.14 ± 0.24	1.00 ± 0.28

The values represent the average value with the standard deviation of 11 trained cup-testers.

**Table 3 metabolites-09-00214-t003:** Top metabolites of coffee fruits and leaf with optimal characteristics (high heritability and strong correlation) linked to sensory traits.

Sensory Attribute	HR *m/z*	Adduct Ion	Name	Tissue	Heritability	SCC	*p*-Value
Acidity	963.4134	n.a.	Unknown	Fruits	0.96	0.667	4.98 × 10^-2^
Aftertaste	940.2271	[M+Na]^+^	2,4-Decadienoyl-CoA	Fruits	0.881	0.754	1.88 × 10^-2^
Aromatic intensity	860.5079	n.a.	Unknown	Fruits	0.999	0.812	7.89 × 10^-3^
Aromatic quality	536.0019	n.a.	Unknown	Fruits	0.656	0.800	9.63 × 10^-3^
Astringency	601.1263	[M+Na]^+^	2,3-trans-proanthocyanidin	Fruits	0.957	0.833	5.27 × 10^-3^
Bitterness	734.4907	n.a.	Unknown	Fruits	0.817	0.740	2.27 × 10^-2^
Body	789.1658	[M+H]^+^	Delphinidin 3-O-glucosyl-5-O-caffeoylglucoside	Fruits	0.949	0.690	3.98 × 10^-2^
Flavor	598.5406	n.a.	Unknown	Fruits	0.998	0.740	2.27 × 10^-2^
Global preference	791.4983	[M+H]^+^	Nonaprenyl diphosphate	Fruits	0.984	0.728	2.61 × 10^-2^
Sourness	521.5618	n.a.	Unknown	Fruits	0.842	0.689	4.00 × 10^-2^
Acidity	473.3339	[M+H]^+^	6-Deoxoteasterone	Leaf	0.643	0.783	1.13 × 10^-2^
Aftertaste	695.4571	[M+Na]^+^	PA(16:0/18:2)	Leaf	0.535	0.737	2.40 × 10^-2^
Aromatic intensity	713.317	n.a.	Unknown	Leaf	0.958	0.803	9.01 × 10^-3^
Aromatic quality	251.2095	[M+Na]^+^	Myristic acid	Leaf	0.994	0.766	1.62 × 10^-2^
Astringency	967.7852	[M]^−^	MGDG(23:0/26:0)	Leaf	0.612	0.666	4.90 × 10^-2^
*Bitterness	473.3339	[M+H]+	6-Deoxoteasterone	Leaf	0.643	−0.857	3.14 × 10^-3^
*Body	463.3451	[M+Na]^+^	4,4-Dimethylzymosterol	Leaf	0.891	−0.689	3.98 × 10^-2^
Flavor	251.2095	[M+Na]^+^	Myristic acid	Leaf	0.994	0.672	4.89 × 10^-2^
Global preference	433.1626	[M+K]^+^	Gibberellin A28	Leaf	0.996	0.677	4.48 × 10^-2^
Sourness	473.3339	[M+H]^+^	6-Deoxoteasterone	Leaf	0.643	0.773	1.47 × 10^-2^

Sensory attributes are presented by alphabetic order and metabolites are showed in decreasing order of heritability level for positive correlation. In some cases, only a negative correlation was detected. HR: High resolution; SCC: Spearman correlation coefficient; n.a.: not applicable; * only negative correlation was detected for this sensory attributes.

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
