# Peer review of "Metabolomic Markers for the Early Selection of Coffea canephora Plants with Desirable Cup Quality Traits"

_metabolites, 2019, doi:10.3390/metabo9100214_

Round 1

Reviewer 1 Report

This is a very interesting and quite comprehensive study of the compounds in coffee and their heritability in Coffea canephora.  The work appears to have been conducted appropriately, but there are some serious problems with the manuscript in its current form.  In addition to the deficiencies described in detail below, there is another serious problem with the paper: the authors claim that their study proves that metabolite profiles in young plants can be used to accelerate the process of plant breeding, yet they have not shown that applying this methodology actually achieves the goal.  They have shown that a metabolomics analysis offers to potential to achieve the goal in a post hoc approach, but have not actually applied the approach and shown that selecting plants with metabolites associated with desirable coffee attributes actually results in the generation of plants capable of producing the desirable traits.  Granted, this will take a bit of time, but it is far too premature to be making such grandiose claims.

Structurally, the major problem with the paper is the inclusion of the correlation analysis among the various sensory attributes (i.e. the data presented in Figure 3).  This information is completely extraneous and the effort required to interpret the data presented in the figure is a huge (and unnecessary) distraction.  This portion of the manuscript should be deleted (along with Figure 3); its absence will not detract in any way from the manuscript.

There are less serious structural problems with the manuscript that need to be corrected.  The description of the correlation analysis in the Materials & Methods (section 3.1.3) should come after the description of the LC-MS analysis (after section 3.4).  In addition, there is information presented in the Results and Discussion section that belong elsewhere (e.g. the first paragraph, section 2.1) contains information that belongs in Materials and Methods and other information that is more of a general discussion and should come after the presentation of results).  The Introduction should be divided into several paragraphs (e.g. there is a natural break in the flow occurring in lines 50 and 60, which is where the paragraph breaks should occur).

Some details are lacking in the Materials and Methods section (e.g. the age of plants when fruit was harvested and the procedure for brewing the beverage, including the amount of ground beans).  There are annoying inconsistencies in the way some of the information is presented as well: the LC gradients are presented in one case in terms of %B and in the other as %A, and results of the offspring-parents regression are not consistently defined as “high” (is 0.4-1 considered “high”?). Furthermore, many readers will not be familiar with “offspring-parents” regression, so the way this analysis was performed needs to be described.

Table 3 has some problems that need to be corrected.  The column headed “Ionization mode” contains not just the ionization mode but also the form of the ion detected (e.g. as M+1 or an adduct of sodium or potassium, or simply not listed).  The column heading needs to be reworded to reflect what is actually presented in the column.  A serious miswording in the footnote to Table 3 also needs to be corrected: “only negative correlation was detected for this metabolites” should be worded “only negative correlations were detected for these sensory attributes”.

Finally, there is extensive edition of English language required.  The authors might benefit from hiring an editing service because the amount of editing required is extensive.  I normally note suggested edits in the manuscripts I review, but the number of changes is far too excessive in this case.

In summary, the study is interesting and the results will be of interest to a fairly wide audience, but the paper requires substantial revision before it can be considered acceptable for publication.

Author Response

Dear Reviewer, thank you very much for your comments and the positive reception of our paper. Following, we give point-by-point answers:

"Structurally, the major problem with the paper is the inclusion of the correlation analysis among the various sensory attributes (i.e. the data presented in Figure 3).  This information is completely extraneous and the effort required to interpret the data presented in the figure is a huge (and unnecessary) distraction.  This portion of the manuscript should be deleted (along with Figure 3); its absence will not detract in any way from the manuscript."

RGB/RW: We believe that the correlation analysis of sensory attributes is highly relevant because it demonstrates that characteristics may be related (genetically/ chemically/ ..) to each other. Therefore, e.g. in breeding programs, they cannot be optimised independently.

"There are less serious structural problems with the manuscript that need to be corrected. The description of the correlation analysis in the Materials & Methods (section 3.1.3) should come after the description of the LC-MS analysis (after section 3.4)"

RGB/RW: We agree and moved the statistical methods below the experimental methods.

"In addition, there is information presented in the Results and Discussion section that belong elsewhere (e.g. the first paragraph, section 2.1) contains information that belongs in Materials and Methods and other information that is more of a general discussion and should come after the presentation of results)."

RGB/RW: We only remind the (quick) reader briefly on the general experimental strategy; the full methods are described in the Materials and Methods section.

"The Introduction should be divided into several paragraphs (e.g. there is a natural break in the flow occurring in lines 50 and 60, which is where the paragraph breaks should occur)."

RGB/RW: Thank you for this suggestion. We agree and added a paragraph break.

"Some details are lacking in the Materials and Methods section (e.g. the age of plants when fruit was harvested and the procedure for brewing the beverage, including the amount of ground beans)"

RGB/RW: We completed the information in the Materials and Metods section: In section 3.1.1, the age of F1 plants at first sampling is given, as well as the age of plants at the second sampling of coffee fruits.

The coffee beverage tasting is now described in more detail in section 3.2.1 ("Coffee beans processing and sensory analysis of coffee beverage").

"There are annoying inconsistencies in the way some of the information is presented as well: the LC gradients are presented in one case in terms of %B and in the other as %A,"

RGB/RW: We unified the presentation of the conditions, to simplify the reading..

"and results of the offspring-parents regression are not consistently defined as “high” (is 0.4-1 considered “high”?)."

RGB/RW: We added a definition for low, moderate and high heritability levels, and applied them for the complete text

"Furthermore, many readers will not be familiar with “offspring-parents” regression, so the way this analysis was performed needs to be described."

RGB/RW: We included a description of the applicability of “offspring-parents regression analysis” in section 3.1.3

"Table 3 has some problems that need to be corrected. The column headed "Ionisation mode" contains not just the ionisation mode but also the form of the ion detected (e.g. as M+1 or an adduct of sodium or potassium, or simply not listed)."

RGB/RW: We added "n.a." (not applicable) for unknown compound adducts.

"The column heading needs to be reworded to reflect what is actually presented in the column."

RGB/RW: "Ionisation mode" was changed to "adduct ion".

"A serious miswording in the footnote to Table 3 also needs to be corrected: “only negative correlation was detected for this metabolites” should be worded “only negative correlations were detected for these sensory attributes”."

RGB/RW: Thanks for this observation, we corrected this as suggested.

"Finally, there is extensive edition of English language required.  The authors might benefit from hiring an editing service because the amount of editing required is extensive.  I normally note suggested edits in the manuscripts I review, but the number of changes is far too excessive in this case."

RGB/RW: We performed additional proofreading and corrected various typos and grammar errors. We are confident that the current version is suitable for publication.

Reviewer 2 Report

It is a very interesting manuscript concerning the application of metabolomics towards the early selection of Coffea canephora plants along breeding processes, as early as 6 months in comparison to the almost 3 years the conventional process based on the fruit’s analysis. In general, the manuscript is generally well writhed, the logic deductions based on observation are well conducted. The conclusions are in general well supported.  Some minor comments were indicated to authors.

Author Response

RGB/RW: Thank you very much for your positive feedback. We performed minor changes according to the comments of all three reviewers and revised the English.

Reviewer 3 Report

The manuscript from Gamboa-Becerra and colleagues describes a very interesting approach aimed to establish some relationship between the metabolic profile of different Coffea canephora plants and the organoleptic quality of the coffee cups. An LC-MS/MS-based un-targeted metabolic analysis was performed in order to identify and quantify the metabolites present in fruits and leaves of 120 coffee plants.

Achieved results are of great interest and suggest the possibility of selecting the different cultivars on the basis of metabolomic markers, in order to obtain coffee cups with a better taste.

However two main problems in the manuscript that have to be solved to make it suitable for publication:

1) The authors developed mass spectrometry method for metabolite identification, based on the use of two different instruments equipped with an Ion Trap and a Q-TOF analyser, respectively. Although this approach seems appropriate to perform qualitative analyses and metabolite identification, it provides poorly accurate quantitative results. I suggest the authors to perform some MRM-based analyses (they could be carried out using a q-tof instrument) to at least confirm their quantitative data.

2) In the manuscript there are no information concerning the sensory analyses performed on coffee cups. As these analyses are critical in the aims of that research, as much as possible information on this aspect has to be provided.

Author Response

Dear Reviewer, thank you very much for your evaluation. Following we will reply to your comments point-by-point:

1) The authors developed mass spectrometry method for metabolite identification, based on the use of two different instruments equipped with an Ion Trap and a Q-TOF analyser, respectively. Although this approach seems appropriate to perform qualitative analyses and metabolite identification, it provides poorly accurate quantitative results. I suggest the authors to perform some MRM-based analyses (they could be carried out using a q-tof instrument) to at least confirm their quantitative data.

RGB/RW: MRM analyses provide highly accurate quantitative data, but the number of mass traces that can be analysed and the sample throughput are limited. Further, the signals of interest have to be defined before the LC-MSn analysis. Consequently, MRM is not suitable for exploratory studies/ undirected metabolomics. In this article, we provide the proof-of-concept, and the data are robust enough to support our claims. For future projects with a defined target (e.g. reduction of coffee bitterness), indeed, MRM analyses would be the right choice.

2) In the manuscript there are no information concerning the sensory analyses performed on coffee cups. As these analyses are critical in the aims of that research, as much as possible information on this aspect has to be provided.

RGB/RW: We extended the information about coffee beverage tasting in section 3.2.1 ("Coffee beans processing and sensory analysis of coffee beverage")